# Predicting the diversity of photosynthetic light-harvesting using thermodynamics and machine learning

**Callum Gray** [ID][1,2]*, **Samir Chitnavis**[1,2], **Tamara Buja**[1,2], **Christopher D. P. Duffy**[1,2]*

**1** Digital Environment Research Institute, Queen Mary University of London, London, United Kingdom,
**2** School of Biological and Behavioural Sciences, Queen Mary University of London, London, United Kingdom

* callum.gray@qmul.ac.uk (CG); c.duffy@qmul.ac.uk (CDPD)

**Data availability statement:** Code used in this paper is available at

## Abstract

Oxygenic photosynthesis is responsible for nearly all biomass production on Earth, and may have been a prerequisite for establishing a complex biosphere rich in multicellular life. Life on Earth has evolved to perform photosynthesis in a wide range of light environments, but with a common basic architecture of a light-harvesting *antenna* system coupled to a photochemical reaction centre. Using a generalized thermodynamic model of light-harvesting, coupled with an evolutionary algorithm, we predict the type of light-harvesting structures that might evolve in light of different intensities and spectral profiles. We reproduce qualitatively the pigment composition, linear absorption profile and structural topology of the antenna systems of multiple types of oxygenic photoautotrophs, suggesting that the same physical principles underlie the development of distinct antenna structures in various light environments. Finally we apply our model to representative light environments that would exist on Earth-like exoplanets, predicting that both oxygenic and anoxygenic photosynthesis could evolve around low mass stars, though the latter would seem to work better around the coolest M-dwarfs. We see this as an interesting first step toward a general evolutionary model of basic biological processes and proof that it is meaningful to hypothesize on the nature of biology beyond Earth.

## Author summary

Photosynthesis is responsible for supplying most of the energy for complex life on Earth. While there are multiple types of photosynthesis, they all have a common feature: an *antenna* system which exists to absorb light and transfer energy to a *reaction centre* where chemical energy is produced. The antennae of photosynthetic organisms have evolved differently to capture more light based on the kind of light they receive. We have developed a physical model of these antenna systems, along with a machine learning algorithm which is given a light input and evolves the antennae to better capture

https://github.com/QMUL-DuffyLab/gala and the data used in preparation is available at https://doi.org/10.5281/zenodo.14514090.

**Funding:** CDPD and CG would like to acknowledge support from the Leverhulme Trust (RPG-2023-096) for direct funding of this project. CDPD and SC would like to acknowledge support from the BBSRC London Interdisciplinary Biosciences PhD Consortium (LIDo). The funders had no role in study design, data collection and analysis, decision to publish, or preparation of the manuscript.

**Competing interests:** The authors have declared that no competing interests exist.

that light. We apply our algorithm to various kinds of light found in different places on Earth, finding that it predicts antennae similar to those found in real organisms. This suggests that common physical principles govern the development of diverse biological structures. We note that photosynthesis presents one of the clearest signals of life (a *biosignature*) on Earth that can be observed from space, called the *vegetation red edge*. Future astronomical missions will search for these biosignatures on exoplanets. We therefore apply our algorithm to other kinds of starlight, and make predictions about what kinds of photosynthesis might evolve on other planets.

## Introduction

With the discovery of nearly 100 potentially-habitable exoplanets [1], the question of whether they do harbour life and what it might be like warrants serious consideration [2]. In which case, how do we meaningfully *hypothesize* (and not merely speculate) on the forms life might take in environments very different to Earth? Regardless of environment, evolution must be constrained by the laws of physics and chemistry; the question is then whether it is possible to develop a model that predicts the type of biological processes that may evolve under a given set of external conditions. Such a model would have be *necessarily* broad and coarse-grained, presupposing no (or very little) fine molecular detail, but still capturing the essential physics of basic biological functions. In essence the problem becomes one of defining elementary biological processes and determining the basic thermodynamic constraints on their function.

The first assumption we make is that life, wherever it evolves, will require an energy source. A common assumption in astrobiology research is that this will be the parent star, with the irradiant flux on the plant surface coupled to the biosphere by some form of photosynthesis [3–7]. In its most basic form, this involves using absorbed light energy to oxidize some chemical substrate to produce reducing agents and chemical energy which are then used in generating biomass. Of course, photosynthesis is certainly not the only mechanism of biomass production, with *chemo*-synthetic organisms oxidizing high-energy compounds from geochemical sources [8,9]. Indeed, recent work has suggested that the last universal common ancestor (LUCA) on Earth was not photosynthetic [10,11]. Regardless, in this work we will exclusively consider photosynthesis. Firstly, stellar light is a universal energy and one that will remain relatively constant over the life of the star. Secondly, the light environment on *known* exoplanets is relatively well-characterized, while conditions on the planets' surface (beyond basic parameters such as temperature and planet density) are not. Thirdly, the presence of photosynthesis on an exoplanet may present the best chances of remote detection. While chemo-autotrophs may have been the first forms of life on Earth, our biosphere (at least on the surface) is dominated by photo-autotrophs and these have left two characteristic signatures: an oxygenated atmosphere and planet surface covered in coloured pigment molecules such as chlorophyll. The presence of oxygen in an exoplanet atmosphere could be detectable as absorption lines in the light passing through its atmosphere as it transits, making it a priority bio-signature [12–14]. On Earth, the global coverage of chlorophyll generates the *vegetation red edge* (VRE) at $\sim$ 700-750 nm in the surface-integrated reflectance spectrum (first observed by the Galileo probe[15]). The potential of measuring a similar reflectance edge is also widely discussed in th context of upcoming missions which will directly image the surface of nearby exoplanets [5,16,17]. We therefore argue not that all biospheres *must* be founded upon some form of photosynthesis, but that any biosphere in general could be, and that photosynthesis represents a promising candidate for the detection of extraterrestrial life if it exists.

The second assumption we make is that the molecular details of photosynthesis will depend on the intensity and spectral quality of incident light. On Earth almost all of the primary production is via *oxygenic* photosynthesis, in which light is used to strip electrons from chlorophyll *a* to produce energy and chemical reductants [18]. The electrons are replenished by the oxidation of water. The redox chemistry of this last step presents a second (energetic) constraint: photons of wavelength longer than about 750 nm are insufficiently energetic to power water oxidation, a widely discussed phenomenon known as the *red limit* of oxygenic photosynthesis [19,20]. While larger and warmer stars such as our Sun emit strongly in the visible range (400-700 nm), providing a significant amount of usable light for oxygenic photosynthesis, cooler and smaller M-dwarf stars have limited emission in this range and higher emission at longer wavelengths, with their absorption maxima reaching out to around 1 $\mu$m. At first glance, then, oxygenic photosynthesis would appear to be unfavourable around such stars. This would appear to be a disappointing conclusion, since M-dwarfs are the most common and longest-lived stars in the universe [21] and have a high occurrence rate of Earth-like planets in their habitable zones [22]. Indeed, well-known, nearby candidate systems such as Trappist-1 [23] and Proxima Centauri [24] orbit such stars. Or course, there is the possibility of *anoxygenic* photosynthesis, which involves photo-oxidation of compounds such as sulphides, hydrogen, iron, etc. and uses much redder (800-900 nm) light [25]. Of course on Earth such organisms are single cell and responsible for less than 1% of the primary production of the biosphere (though this can locally increase to $\sim$ 30% in ancient, sulphide rich lakes [26]). Moreover, it has been proposed that an oxygenated atmosphere is a necessary prerequisite for the evolution of widespread multicellular life (though there is considerable debate [27–31]), meaning a purely anoxygenic biosphere may not produce the widespread land-based vegetation needed to generate a detectable VRE. Fortunately, recent experiments have shown that various oxygenic cyanobacteria, algae, moss and non-vascular plants are able to grow under simulated M-dwarf light [6,32], though vascular plants fare less well, showing signs of the stress-response known as *shade avoidance syndrome* [6]. These are extremely encouraging results as they prove that oxygenic photosynthesis around M-dwarf stars is not automatically precluded by their paltry emission above 700 nm. That said, this does not mean oxygenic photosynthesis would evolve in such conditions, and our recent theoretical work indicated that oxygenic organisms would require larger and more complicated light-harvesting structures than they do on Earth [7].

The third basic assumption we make is that photosynthetic processes in general will evolve some form of *antenna* structure. Essentially all of the various photoautotrophs on Earth adopt this approach, in which a small number of photochemical *reaction centres* (RC) are supplied energy by a much larger antenna system [33]. While there is significant diversity in the structural details of the various antennae which have evolved in different light environments, they function according to the same principles: A large, modular assembly of pigment-protein complexes absorb incident light and very efficiently transfer the resulting excitation energy to the RCs. That said, they tend not to absorb light indiscriminately across the visible spectrum, but rather selectively absorb specific (and sometimes rather narrow) wavelength bands. Previous theoretical work has suggested that selective light-harvesting structures may have developed to balance capturing light against the metabolic cost of synthesising multiple pigment types [34], to absorb light preferentially at the maximum incident photon flux [4,35], or to mitigate fluctuations in the light level in order to maintain the average power throughput of the antenna [36]. In this work we consider a different potential organising principle: balancing photon flux capture, the thermodynamics of energy transfer, spectral overlap of between antenna components, and the metabolic burden of building and maintaining a large, complex antenna. The first consideration, maximizing photon flux capture, is obvious. We

assume a photoautotroph, requiring energy and subject to no other constraints, would evolve an antenna which intercepted the largest amount of incident flux. This would imply a huge antenna containing multiple pigment types to cover as much of the incident spectrum as possible. The thermodynamic constraint reflects the fact that light-harvesting, intercepting photons over a large area and concentrating the resulting energy into a small RC, involves an unfavorable reduction in entropy ($\Delta S < 0$) [37–39]. To counteract this, light-harvesting generally also involves a reduction in *enthalpy* ($\Delta H < 0$) so that the total change in free energy, $\Delta F = \Delta H - T\Delta S$, is negative (or at least not prohibitively positive). In, for example, Photosystem II (PSII) of vascular plants it is sufficient to have the RC slightly lower in energy than the surrounding (and somewhat iso-energetic) antenna [40,41]. For cyanobacteria, the antenna (known as the phycobilisome) could be described as 'funnel-like', with higher-energy pigments located further away from the RC. Still, an antenna cannot be arbitrarily large and in our previous work we showed that increasing antenna size eventually leads to diminishing returns in light-harvesting as the entropic barrier becomes harder and harder to overcome [7]. 'Spectral overlap' essentially means that the antenna and RC must be energetically resonant otherwise energy transfer would violate the conservation of energy. This is also true for the different pigment types/domains in the antenna. In practice this means one cannot have an arbitrarily steep funnel [7]. Finally, we assume that having a huge antenna will place some metabolic burden on the organism. This could be in the direct sense of having to divert a fraction of photosynthetic output to build and maintain the antenna system, or in a more abstract sense of taking up valuable space in the cell. In this work we create a general antenna model that captures these constraints. We then use a genetic/evolutionary algorithm to optimize this antenna to different light-environments.

## Methods

We construct a thermodynamic model of a general light-harvesting antenna. The model parameters (discussed below) are related to overall structural features such as size, shape and pigment composition. The 'input' for a given antenna is an incident spectral flux of light and the model outputs the overall *electron output rate*, $\nu_e$, and the absolute quantum efficiency, $\phi$, of the antenna. We then generate random populations of these antennae and apply an evolutionary algorithm to mimic the processes of selection, reproduction and mutation in order to produce antennae better optimised to the light environments they are subjected to.

### Antenna model

**Thermodynamics of light-harvesting.** We use a relatively simple kinetic model of photon absorption and energy transfer within a given light-harvesting structure based on that previously reported in [7], with a diagram of an example antenna shown in Fig 1A.

The model contains one generalized RC containing $N_r$ pigments plus exactly one trap state $N_t = 1$ which converts an exciton produced by absorption of a photon into a charge-separated state. The rest of the antenna is composed of $n_b$ identical branches, each made up of $n_s$ light-harvesting complexes (*LHCs*) which we refer to as *subunits*; these in turn contain $n_i$ identical pigments chosen from the set shown in Fig 1D and described in further detail in Tables A and B of the S1 Text.

We consider five main processes occurring in the antenna: photon absorption, energy dissipation in the antenna, energy transfer, energy trapping by the RC, and electron output. We do not consider dissipative processes within the RC (such as non-radiative charge recombination), since we are primarily concerned here with the effect of antenna evolution. We do not present a rigorous treatment of charge-separation and transfer in the RC at all

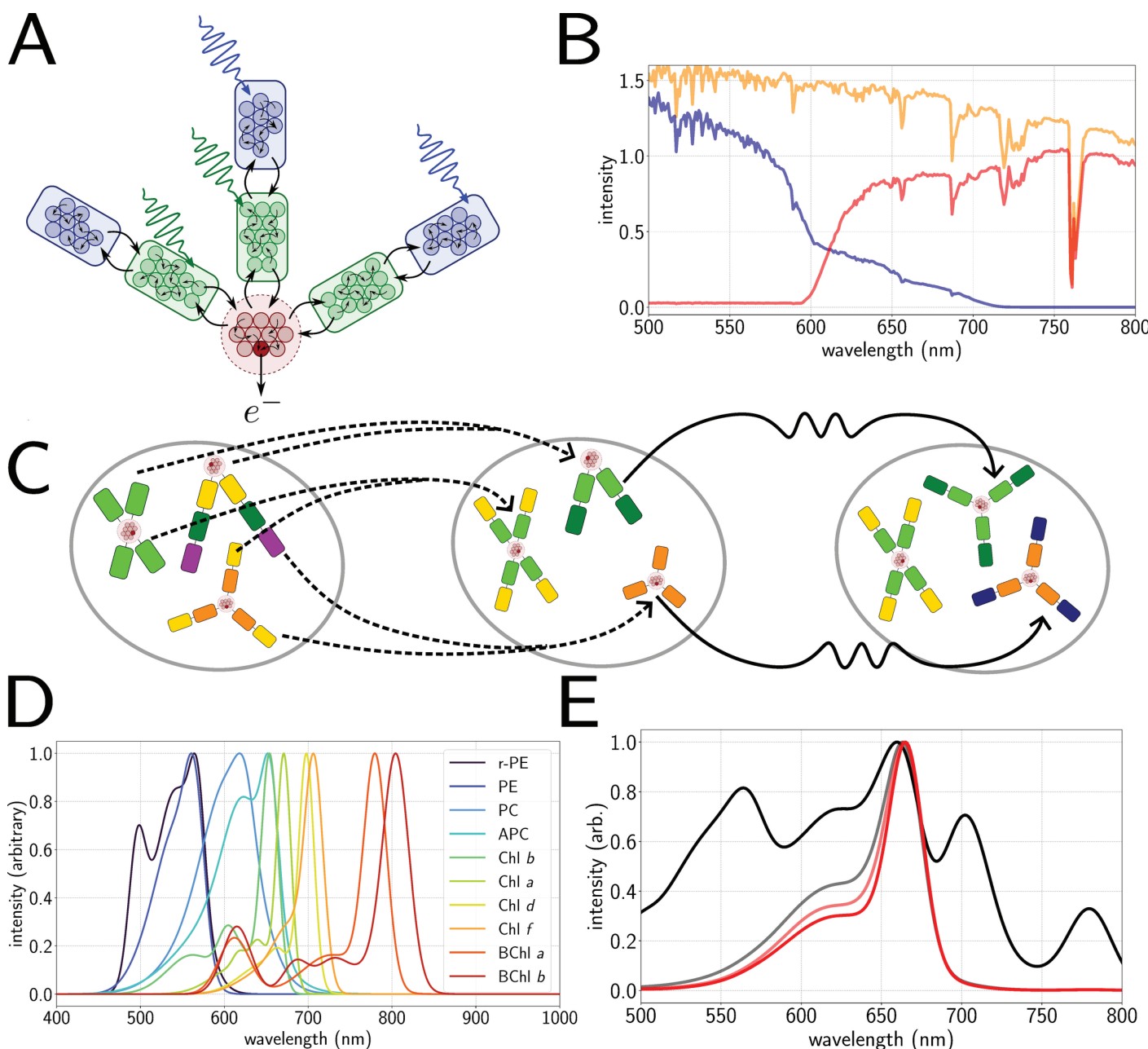

**Fig 1. A.) Thermodynamic model of antenna. Rounded squares represent antenna subunits with the small circles representing pigments.** Colour (green, blue) crudely represents absorption wavelength. Light is absorbed by each sub-unit and energy is transferred between them (black arrows), ending at the reaction centre (red) which traps (dark red) the energy and outputs electrons. B.) Altering the light environment from full sunlight (yellow) to deep water (blue) or shaded (red) via filters. C.) Genetic algorithm. Pairs of parents are selected (dashed lines) from the initial population to reproduce (left to middle), after which a mutation step is applied to members of the population at random (middle to right) to form the next generation. D.) Fitted absorption lineshapes of the available pigments for the genetic algorithm to choose from [33,42,43]. E.) Example of evolution of the average absorption spectrum from initial population (black) to a converged population (red) under red light illumination.

and so we assume that dissipative processes simply renormalize (and are already accounted for in) the rate constant for electron output. We make the assumption that an RC would evolve to minimize such waste channels as they ultimately defeat the purpose of the RC. A

more detailed model of the combined antenna-RC supersystem will be the focus of our next work.

Photon absorption is described by the excitation rate $\gamma_i$ as

$$\gamma_i = \int_0^\infty \frac{\lambda}{hc} f_p(\lambda) A_i(\lambda) \, d\lambda \tag{1}$$

where $f_p(\lambda)$ is the incident spectrum (which can be one representing an Earth-like light environment or an extrasolar spectrum, with examples shown in Fig 1B) and $A_i(\lambda)$ is the absorption spectrum of the pigments in subunit $i$. $\lambda/hc$ ensures that the resulting quantity has dimensions of photons $m^{-2}$ $s^{-1}$.

Energy dissipation in the antenna is assumed to be due to realtively fast processes such as non-radiative excitation decay and fluorescence, which we combine these into one rate $k_{\text{diss}}$.

Considering the kinetics of energy trapping in and electron output from the reaction centre requires us to describe explicitly whether the trap is already closed (occupied) or open (unoccupied), since this affects which processes are possible. The relevant equations are

$$\left[ \frac{d}{dt} P_{n_r,0}(t) \right]_{\text{trap}} = -n_r k_{\text{trap}} P_{n_r,0}(t) \tag{2}$$

$$\left[ \frac{d}{dt} P_{n_r,1}(t) \right]_{\text{trap}} = -k_{\text{out}} P_{n_r,1}(t) + (n_r + 1) k_{\text{trap}} P_{n_r+1,0}(t) \tag{3}$$

for the open and closed traps respectively. Not that here we define *trapping* as the process of an exciton localized on the RC undergoing charge separation. In the photosynthetic literature 'trapping' can also include the migration of the exciton to the RC, which here is accounted for in our energy transfer terms. In real photosynthetic systems the kinetics of trapping involves multiple steps involving progressively longer timescales and larger distances. The initial charge separation steps are fast (order of 10-100 ps), while later steps involving reduction of electron carriers can be much slower ($\mu$-ms). In our model we class the very earliest charge separation steps as trapping, since it is at this point the energy transfer gives way to electron transfer, with the latter not being rigorously treated in our model. We therefore define $k_{\text{trap}}^{-1} \sim 5$ ps for two reasons. Firstly, it is the approximate timescale of the formation of the radical pair states in PSII [41,44] and secondly, if trapping is much slower then the back transfer of excitation into the antenna becomes the favourable process and the entire system ceases to work effectively.

The timescale for acceptor reduction, $k_{\text{out}}^{-1}$ is *much* slower than the other timescales in the model. We will choose $k_{\text{out}}^{-1} \sim 10$ ms as this is approximately the timescale of the $H_2O$ oxidation-quinone reduction cycle of PSII [45]). As previously stated, severe oversimplification of a complex system. There is even some evidence that the instantaneous state of the RC may also modulate exciton migration in the antenna [46]. However, as previously stated, this is left for our follow-up work.

Finally we consider energy transfer rates between subunits, beginning with a fundamental hopping rate $k_{\text{hop}}$ which sets an overall timescale. This will depend to some extent on details of the antenna which we deliberately do not model, but we can say in general that $k_{\text{hop}} >> k_{\text{diss}}$ in order for light-harvesting to work effectively; we use a fundamental rate $k_{\text{hop}} = 10$ps in this work, which is the typical timescale of energy transfer between subunits in real photosystems [40]. This base hopping rate is then modified by the overlap of the spectral lineshapes of

the initial and final subunits and the free energy change using,

$$K_{n_i,n_j}^{n_i-1,n_j+1} = k_{\text{hop}} n_i A_{ij} \, \rho \left( E_i - E_j \right) f\left( \Delta F_{n_i,n_j}^{n_i-1,n_j+1} \right) \tag{4}$$

for hopping from subunit $i \to j$ with initial subunit populations as subscript and final as superscript. $A_{ij}$ is simply an adjacency term equal to 1 if the two subunits are connected and 0 otherwise. $\rho(E_i - E_j)$ is the *density of states* which enforces energy conservation on the process; energy transfer is only possible between subunits if the pigments in the subunits are at resonance, and the degree of resonance is approximately captured by the overlap integral,

$$\rho \left( E_i - E_j \right) \sim \int_0^\infty d\lambda \tilde{F}_i \left( \lambda; \lambda_i^p, w_i \right) \tilde{A}_j \left( \lambda; \lambda_j^p, w_j \right) \tag{5}$$

where $\tilde{F}_i$ and $\tilde{A}_j$ are the normalized fluoresence and absorption lineshapes of the donor $i$ and acceptor $j$ respectively. Finally $f\left( \Delta F_{n_i,n_j}^{n_i-1,n_j+1} \right)$ encodes the thermodynamics of energy transfer via the free energy change. At equilibrium the rates of forward and backward energy transfer between any pair of subunits must satisfy the *detailed balance* condition:

$$\frac{K_{n_i,n_j}^{n_i-1,n_j+1}}{K_{n_i-1,n_j+1}^{n_i,n_j}} = \left( \frac{n_i}{n_j + 1} \right) \exp\left( -\frac{\Delta F_{n_i,n_j}^{n_i-1,n_j+1}}{k_B T} \right) \tag{6}$$

$\Delta F_{n_i,n_j}^{n_i-1,n_j+1}$ is the Helmholtz free energy change associated with the transfer process,

$$\Delta F_{n_i,n_j}^{n_i-1,n_j+1} = \Delta H_{n_i,n_j}^{n_i-1,n_j+1} - T\Delta S_{n_i,n_j}^{n_i-1,n_j+1} \tag{7}$$

The first term here $\Delta H$ represents the enthalpy change, i.e. the difference in internal energy of the system before and after the transfer process,

$$\begin{aligned} \Delta H_{n_i,n_j}^{n_i-1,n_j+1} &= (n_i - 1) E_i + \left( n_j + 1 \right) E_j - n_i E_i - n_j E_j \\ &= E_j - E_i \\ &= hc \left( \frac{1}{\lambda_j^p} - \frac{1}{\lambda_i^p} \right) \end{aligned} \tag{8}$$

where $E_{i,j}$ are the excitation energy of the pigments in sub-unit $i, j$, which is inversely proportional to $\lambda_{i,j}^p$, the peak absorption wavelengths.

The second term $\Delta S$ is the entropy change of the process. If we have two coupled LHCs with $N_i$ and $N_j$ identical pigments respectively, of which $n_i \leq N_i$ and $n_j \leq N_j$ are excited, the combined entropy of the two is

$$S_{n_i,n_j} = k_B \ln \left[ W(n_i, N_i) W(n_j, N_j) \right] \tag{9}$$

where $W(n_i, N_i)$ is the multiplicity,

$$W(n_i, N_i) = \frac{N_i!}{n_i! \, (N_i - n_i)!} \tag{10}$$

If one excitation is transferred from $N_i$ to $N_j$ then the entropy change is,

$$\Delta S_{n_i,n_j}^{n_i-1,n_j+1} = k_B \ln\left( \frac{W(n_i-1,N_i)\,W(n_j+1,N_j)}{W(n_i,N_i)\,W(n_j,N_j)} \right) \tag{11}$$

Implicit in this is the assumption that excitons equilibrate *within* each sub-unit much faster than the typical timescale of energy transfer *between* them, so that they can sample all the available states (ergodicity). We have previously shown that this is true up to around $N_p = 100$ [7], considering the typical equilibration times and hopping rates of real photosystems, and hence set a maximum subunit size of 100 pigments throughout this work; in practice, this limit is never reached.

$\Delta F$ is antisymmetric with respect to reversal of the the transfer process,

$$\Delta F_{n_i,n_j}^{n_i-1,n_j+1} = -\Delta F_{n_i-1,n_j+1}^{n_i,n_j} \tag{12}$$

and hence, for non-identical subunits, there will be a thermodynamically favoured direction of energy transfer. This is encoded into the function

$$f\left(\Delta F_{n_i,n_j}^{n_i-1,n_j+1}\right) = \begin{cases} 1 \text{ for } \Delta F_{n_i,n_j}^{n_i-1,n_j+1} \le 0 \\ \exp\left(-\beta\Delta F_{n_i,n_j}^{n_i-1,n_j+1}\right) \text{ for } \Delta F_{n_i,n_j}^{n_i-1,n_j+1} > 0 \end{cases} \tag{13}$$

where $\beta = 1/k_B T$ is the inverse thermodynamic temperature. If the forward transfer process, $K_{n_i,n_j}^{n_i-1,n_j+1}$ represents a decrease in free energy ($\Delta F < 0$), then the backward rate is unfavourable ($\Delta F > 0$) and subject to a multiplicative *Boltzmann penalty*, and vice-versa.

Finally, to construct the set of time-dependent rates which we will solve for, we assume the *single excitation regime*:

$$n_i = 0, 1 \tag{14}$$

$$n_r = 0, 1 \tag{15}$$

$$\langle n_r \rangle + \sum_i \langle n_i \rangle \le 1 \tag{16}$$

In words, the entire antenna-plus-RC (hereafter 'photosystem') can contain a maximum of one excitation at any time, though the trap can be either open or closed. This is reasonable because $k_{\text{diss}}$ and $k_{\text{trap}}$ are much faster than $\gamma_i$, even in bright light. This assumption allows us to neglect multi-excitation terms in our model and if we adopt notation,

$$P_{0,0,n_t} = P\left(n_1 = 0, n_2 = 0, \ldots, n_i = 0, n_r = 0, n_t\right) \tag{17}$$

$$P_{1_i,0,n_t} = P\left(n_1 = 0, n_2 = 0, \ldots, n_i = 1, n_r = 0, n_t\right) \tag{18}$$

$$P_{0,1,n_t} = P\left(n_1 = 0, n_2 = 0, \ldots, n_i = 1, n_r = 1, n_t\right) \tag{19}$$

and so on, then the equations of motion reduce to,

$$\frac{d}{dt}P_{0,0,0}(t) = -\sum_i \gamma_i N_i(1-n_i)P_{0,0,0}(t)$$
$$+ k_{\text{diss}}\left( P_{0,1,0}(t) + \sum_i P_{1_i,0,0}(t) \right) + k_{\text{out}}P_{0,0,1}(t) \tag{20}$$

$$\frac{d}{dt}P_{0,0,1}(t) = -\left(\sum_i \gamma_i N_i(1-n_i) + k_{\text{out}}\right)P_{0,0,1}(t)$$
$$+ k_{\text{diss}}\left(P_{0,1,1}(t) + \sum_i P_{1_i,0,1}(t)\right) + k_{\text{trap}}P_{0,1,0}(t) \tag{21}$$

$$\frac{d}{dt}P_{1_i,0,0}(t) = -\left(k_{\text{diss}} + k_{i\to r} + \sum_j k_{i\to j}\right)P_{1_i,0,0}(t)$$
$$+ \gamma_i N_i(1-n_i)P_{0,0,0}(t) + k_{r\to i}P_{0,1,0}(t) + \sum_j k_{j\to i}P_{1_j,0,0}(t)$$
$$+ k_{\text{out}}P_{1_i,0,1}(t) \tag{22}$$

$$\frac{d}{dt}P_{1_i,0,1}(t) = -\left(k_{\text{diss}} + k_{\text{out}} + k_{i\to r} + \sum_j k_{i\to j}\right)P_{1_i,0,1}(t)$$
$$+ \gamma_i N_i(1-n_i)P_{0,0,1}(t) + k_{r\to i}P_{0,1,1}(t) + \sum_j k_{j\to i}P_{1_j,0,1}(t)$$
$$+ k_{\text{trap}}P_{1_i,1,0}(t) \tag{23}$$

$$\frac{d}{dt}P_{0,1,0}(t) = -\left(k_{\text{diss}} + k_{\text{trap}} + \sum_i k_{r\to i}\right)P_{0,1,0}(t)$$
$$+ \sum_i k_{i\to r}P_{1_i,0,0}(t) + k_{\text{out}}P_{0,1,1}(t) \tag{24}$$

$$\frac{d}{dt}P_{0,1,1}(t) = -\left(k_{\text{diss}} + k_{\text{out}} + \sum_i k_{r\to i}\right)P_{0,1,1}(t)$$
$$+ \sum_i k_{i\to r}P_{1_i,0,1}(t) \tag{25}$$

where the rate constants are defined as

$$k_{i\to j} \equiv K_{1_i,0_j}^{0_i,0_j}. \tag{26}$$

The thermodynamic quantities also simplify:

$$\Delta H_{1_i,0_j}^{0_i,1_j} \equiv \Delta H_{i\to j} = hc\left(\frac{1}{\lambda_j^p} - \frac{1}{\lambda_i^p}\right) \tag{27}$$

$$\Delta S_{1_i,0_j}^{0_i,1_j} \equiv \Delta S_{i\to j} = k_B \ln\left(\frac{N_j}{N_i}\right) \tag{28}$$

Now that all the relevant parameters and rates have been defined we can numerically solve the equations of motion in the steady state:

$$\frac{d}{dt}P_{n_r,n_t} = 0 \tag{29}$$

subject to the constraints

$$\sum_{n_i,n_r,n_t} P_{n_r,n_t}(t) = 1 \tag{30}$$

$$0 \leq P_{n_r,n_t}(t) \leq 1 \tag{31}$$

to obtain the full set of equilibrium probabilities $\mathcal{P}_{n_r,n_t}$. From these we can obtain the *average occupancies*

$$\langle n_t \rangle = \sum_{n_i,n_r} \mathcal{P}_{n_r,1} \tag{32}$$

$$\langle n_r \rangle = \sum_{n_i,n_r} n_r \left( \mathcal{P}_{n_r,0} + \mathcal{P}_{n_r,1} \right) \tag{33}$$

$$\langle n_i \rangle = \sum_{n_i,n_r} n_i \left( \mathcal{P}_{n_r,0} + \mathcal{P}_{n_r,1} \right) \tag{34}$$

which we then use to calculate two observable quantities. The first is the *electron output rate*,

$$\nu_e = k_{\text{out}} \langle n_t \rangle \tag{35}$$

which can be measured (indirectly) for oxygenic photo-autotrophs using an oxygen electrode. The second is the *absolute antenna quantum efficiency*,

$$\phi_e = \lim_{\gamma_i \to 0} \left[ \frac{\nu_e}{\nu_e + k_{\text{diss}} \left( \langle n_r \rangle + \sum_i \langle n_i \rangle \right)} \right] \tag{36}$$

which is the fraction of captured photons that go on to generate an electron in the limit of low illumination. We note that $\phi_e$ is similar but not directly comparable to $\Phi_{\text{PSII}}$, the PSII quantum efficiency, which is routinely measured using specialized fluorimeters in order to quantify the performance of crop species [47,48], and essentially measures the efficiency of excitation trapping. We use $\nu_e$ and $\phi_e$ to quantify the performance of our generalized photosystems for a given light input.

**Constraints on overall antenna structure.** The possible ways in which a collection of different pigments could be arranged into an antenna represents a intractably vast parameter space. We therefore impose some reasonable constraints on the overall geometry. We assume that the RC lies at the centre of our system with some arbitrary number, $n_b$, of antenna branches radiating outward (see Fig 1**A**). A branch can contain an arbitrary number, $n_s$, of subunits which each contain an arbitrary number, $n_p^i$ of pigments. We assume that (1) all branches are identical and (2) the pigment states in a subunit are all isoenergetic. The first constraint is applied purely to ensure computational tractability and is partially justified by the modular/symmetric nature of many real antenna systems. However, highly asymmetric antennae do exist and this should be considered in future work. The second constraint is a reflection of the assumption of a separation of timescales. Energy equilibrates rapidly (we assume essentially instantaneously) over clusters of closely-packed, chemically-similar pigments, while energy transfer between such clusters is assumed to be slower ($\sim 10$ ps). The subunits of our model are not meant to necessarily reflect entire protein complexes, merely clusters of strongly coupled pigments. Similarly, the number of pigments in a subunit does not necessarily equate to the number of actual molecules, but simply the number of *thermodynamically equivalent states*. If one has a cluster of $N_p^i$ pigments with different energies, $E_i$, then the number of *equivalent* pigment states is

$$n_p^i \sim \left\lceil \sum_{i=1}^{N_p^i} \exp\left( -\frac{E_i}{k_B T} \right) \right\rceil \tag{37}$$

Finally, we set some upper limits on parameters based on computational considerations of memory and simulation time. As mentioned above, we assume $n_p^i \leq 100$ which is justified by our previous work [7]. We also set $n_s \leq 100$ and $n_b \leq 12$, with the latter partially justified by close-packing considerations (you can pack 12 spheres around a central sphere of the same size). However, in practice, none of these limits are ever reached in the evolutionary optimization. Beyond this we impose no other structure on our antenna system, allowing the genetic optimization to choose effective structures. There is nothing to stop the model selecting an entirely isoenergetic antenna or one with entirely random energy landscape (beyond the fact that this is unlikely to work very well).

One should not over-interpret the various antenna structures produced by out model. They are better thought of in terms of pigment/spectral composition and topology rather than strict real-space models.

**Excitation transfer between antenna branches.** It should be possible in general for energy to be transferred between adjacent branches, and we initially allowed for this possibility in our simulations, but found that it has no effect whatsoever on the resulting kinetics; this is because the branches are assumed to be identical, and hence at equilibrium the rates of transfer between them are identically equal as well. In a radially-symmetric system such as ours, the radial and lateral hopping of the excitation are independent and therefore antenna efficiency *cannot* be improved (or decreased) by allowing hopping between branches. After verifying this we disabled energy transfer between branches, and do not consider it in any of the work published here. Note that any consideration of non-identical branches or non-equilibrium interactions within the complex would necessarily require radial transfer between branches to be properly accounted for.

**Different pigment types available in the model.** We use a range of pigment lineshapes which are taken from various organisms in different photosynthetic niches. For each subunit of the antenna the algorithm is free to choose any of the available pigment types when constructing or mutating the antennae. This choice is implemented by taking published absorption and emission lineshapes for each pigment type in solution [33,42,43] and fitting them to a set of Gaussian functions. The available choices are shown in Fig 1D; the individual fits as well as the corresponding fits to fluorescence lineshapes are given in Tables A and B and Figs A–C in S1 Text. These fitted Gaussians are then recalculated for the set of wavelengths given in the input spectra to avoid any interpolation issues and also to allow for future work where it may be necessary to introduce shifting of absorption peaks.

We do not consider excitonic mixing or inhomogenous broadening of these lineshapes since we do not model the microscopic structure of the antenna. Of course, looking at the effects of these mechanisms would be a worthy extension of this work.

## The genetic algorithm

The genetic algorithm used in this work uses the standard selection, crossover and mutation steps [49–51], as schematically represented in Fig 1**C**. The antenna parameters $n_b$, $n_s$, and the full set of $n_p^i$ and pigment types constitute the *genome* of our generalized antenna. A *population* of such genomes is initialized and their *fitness* is evaluated in terms of the overall electron output, $\nu_e$, that the antenna can generate in the pre-specified light environment. The relative fitness of the different members of the population are then used to determine the inherited properties of the next generation, with *crossover* capturing the gene-mixing process of sexual reproduction. Throughout the evolution we track various quantities averaged over the population, including the average absorption spectrum of the antennae, as shown in Fig 1E.

Since we allow the size of the antenna and its pigment composition to vary, we then have genomes of arbitrary length which are themselves composed of a mixture of numerical and categorical variables; the implementation of the crossover and mutation algorithms in particular then require some care. We detail the procedures for each of the operators here.

**Initialisation of the population.**   We allow for two different methods of creating an initial population of antennae. The first, which is used throughout most of this work, represents *in situ* evolution of an organism in a given environment. To represent this we consider an initial population of *proto-antennae* all composed of a single branch with a single subunit containing a random number of a random pigment: $n_b = 1, \; n_s = 1, \; n_p \in [1, 100]$.

Alternatively we can model *contingent* evolution as discussed below in the context of antenna evolution in underwater environments. In this case we provide a template for the genetic algorithm to begin from together with a variability parameter. The starting template can be the best/average outcome of a previous simulation or some biologically rationalized model. The variability parameter takes values between 0 and 1 and denotes the fraction of the population which should be mutated using the mutation algorithm described below before the simulation begins. Setting the variability to 0 then provides an initial population of identical antennae.

**Selection based on fitness.**   The fitness function for which the algorithm optimises is given by

$$F = \nu_e - \chi \left( n_b \sum_{i=1}^{n_s} n_p^i \right) \tag{38}$$

where $\chi$ is a pre-defined cost parameter. $\chi$ represents an abstract metabolic burden associated with building and maintaining a large antenna system, a burden that increases with the size of the antenna. For example, $\chi = 0.02$ would mean that an antenna containing 100 pigments would cost 2 electrons per second, deducted from the total of $\nu_e$ produced by the RC. Obviously a large number of processes could contribute to $\chi$, such as the metabolic cost of synthesising proteins and pigments, the cost of supporting non-photosynthetic tissues, the sacrifice of physical space within the cell, etc. It is not intended to be a quantitative measure, but qualitatively reflects the fact that building the photosynthetic machinery would come at some expense. It most likely differs for different types of organism and we would expect, say, $\chi$ to be higher for vascular plants than for prokaryotes. Practically, $\chi$ prevents the genetic algorithm from selecting huge but very inefficient antenna, as shown in Fig 2A. Consider a situation where one doubles the size of the antenna for an increase in $\nu_e$ of 0.001%. If $\chi = 0$ then this technically reflects an increase in fitness, albeit an extremely marginal one.

Selection on the basis of fitness is performed using stochastic universal sampling via the cumulative probability

$$p_m = \frac{1}{\sum_m F_m} \left( 1 - \exp\left( -\frac{F_m}{F_{\max}} \right) \right) \tag{39}$$

where $F_m$ denotes the fitness of the $m$'th antenna in the population and $F_{\max} = 100 \text{ s}^{-1}$ is the maximum possible fitness. In this way pairs of parents are chosen to reproduce; we use a generational model, where the entire population is replaced after each generation. Note that this represents a high selection pressure, and in particular prevents any antenna whose fitness $F < 0$ from reproducing at all, which we identify with individuals whose antennae are so badly optimised for the light environment that they cannot capture enough energy to survive.

**Crossover.**   The multivariate nature and differing lengths of the genomes in the population make standard procedures such as one- or two-point crossover unfeasible in our case; instead,

we perform *intermediate recombination* [49] on our pairs of parent antennae when constructing new child genomes. This is done using a recombination parameter $d = 0.25$ [52] and the function

$$x_{\text{child}} = x_{p_1} b + x_{p_2} (1 - b) \tag{40}$$

where $x \in n_b, n_s, n_p^i$ and the subscripts $p_1$ and $p_2$ indicating parents 1 and 2 respectively. $b = \text{random}(-d, 1 + d)$ is a random number weighting the contribution of parent 1 versus parent 2, and the subsequent value is rounded to integer for integer parameters.

The pigment type associated with a particular antenna subunit is a categorical variable; in this case, for each subunit, the algorithm chooses randomly from the pigment types of the two parents at that subunit. If the number of subunits of the child exceeds that of both its parents, the pigment type of the final subunit is used instead.

**Mutation.**   Again, due to the non-typical nature of the genomes used, it is necessary to define a bespoke mutation operator for our antennae. In order to reduce the number of introduced hyperparameters and avoid imposing any assumptions about which mutations should be more or less common, we choose to use a single mutation rate $r = 0.05$; when a particular antenna is chosen to mutate, we mutate several of its parameters one by one by drawing new values from a Gaussian distribution. In this way we allow all the relevant characteristics of our antennae to mutate without creating a hierarchy of mutations which presuppose a particular structure or set of biosynthetic pathways. The Gaussian distribution from which we draw new values for a given parameter is centred at the current value of the parameter $\mu_{\text{mutation}}$ and has a width of $\sigma_{\text{mutation}} = 0.1\mu_{\text{mutation}}$; it is truncated where necessary to avoid unphysical values being accepted, for example by setting the number of branches or blocks to zero. The exception to this procedure is again the pigment type of a given subunit; in this case, mutation draws a new pigment type as a random choice from the set of available pigment types with no bias towards those already present in the antenna.

## Results

### Oxygenic photosynthesis in full sunlight

Fig 2 shows the results of simulating antenna evolution using the standard AM1.5 sunlight spectrum [53] for a range of values of the cost parameter $\chi$. This corresponds to very bright illumination, equivalent to a photosynthetically active (400–700 nm) photon flux of over 1000 $\mu$mol m$^2$ s$^{-1}$; note that we neglect the effect of any photoprotective processes such as non-photochemical quenching (NPQ); the precise molecular mechanisms of this process are still under debate [54] and a proper treatment is far beyond the scope of this work. The resulting averaged electron output $\nu_e$ and antenna efficiency $\phi$ in Fig 2B show very high electron output, up to $80 s^{-1}$ at lower cost values where the antennae are larger, and high efficiencies of 70–90%. The averaged antenna absorption spectra shown in Fig 2C have absorption peaks at around 650 nm, 610 nm and 570 nm, corresponding to the cyanobacterial pigments allophycocyanin (APC), phycocyanin (PC) and phycoerythrin (PE) respectively, with differing proportions depending on cost. More detail on pigment composition is available in Table 1 and in the Fig D in S1 Text. We note that for the lowest costs we also start to see small peaks appearing in the average absorption spectra at around 700 and even 800 nm, corresponding to the inclusion of pigments such as Chl *d* and BChl *a*, along with a decrease in the efficiency of the antennae. These pigments do not match the peaks in incident flux and transfer very little energy to the reaction centre, reflecting the fact that as the cost decreases it is less disadvantageous to build a larger, less efficient antenna. Below $\chi = 0.005$ the population becomes even larger and less efficient with more and more unusual features appearing in the absorption

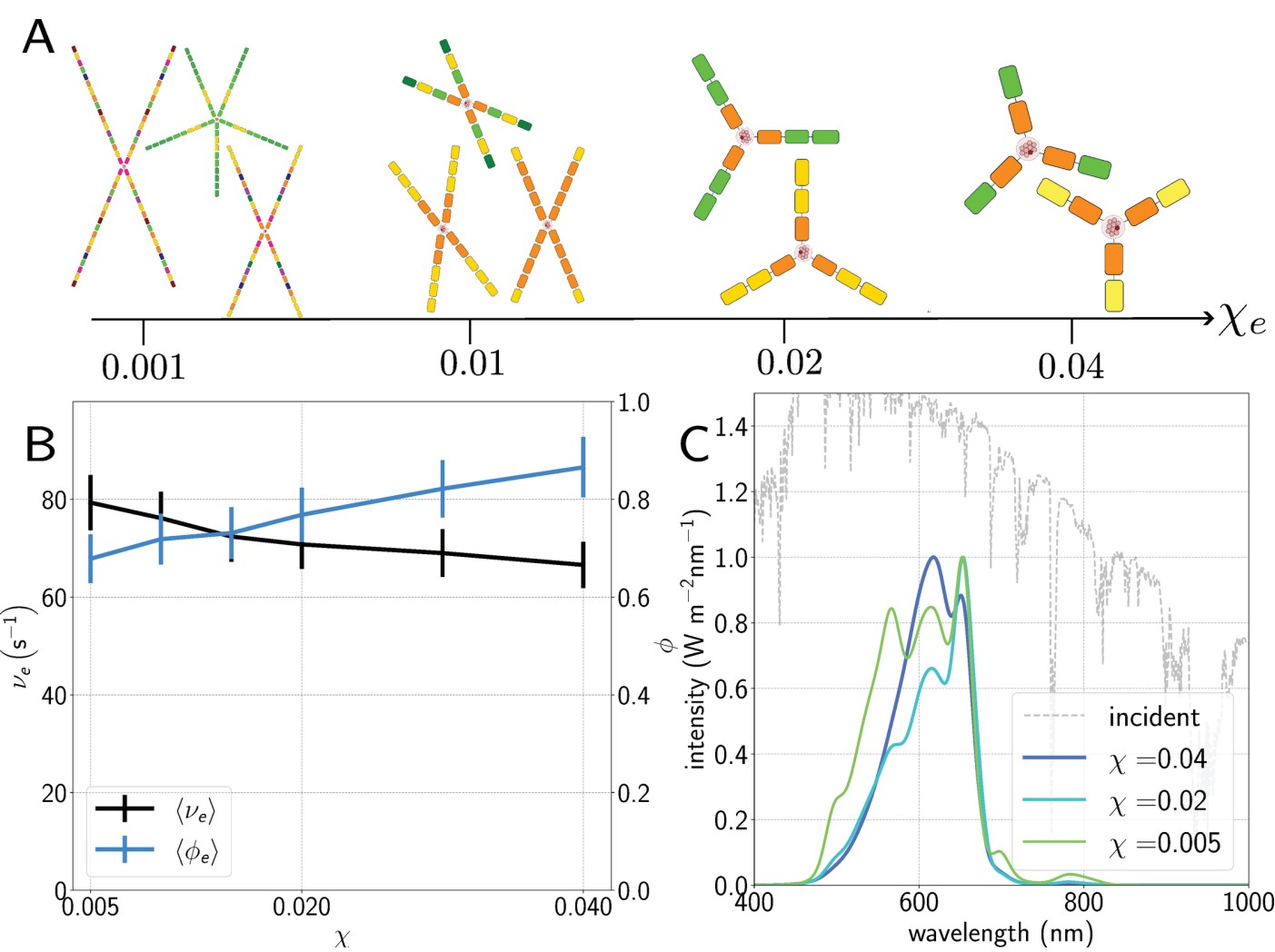

**Fig 2. A.) Pictorial representation of light-harvesting structures evolved for different values of the cost $\chi$.** Different pigment types represented as different colours; the colours themselves are arbitrary. At very low cost, the antennae become extremely large and contain pigments which are useless for light-harvesting; at too high a cost they fail to evolve at all. B.) Average electron output per second $\nu_e$ and antenna efficiency $\phi$ as a function of $\chi$ for antennae evolved under standard AM1.5 spectrum [53] C.) Average absorption spectrum of total population as a function of $\chi$, showing a longer tail and more featured spectrum as the cost decreases. The y-axis here and throughout denotes the intensity of the incident light (grey dashed line) rather than the absorption curves, which are simply normalised by setting their maximum to 1 for readability.

spectra, whereas above $\chi = 0.04$ it can become difficult to evolve any population at all depending on light environment. Hence for the rest of this work we apply bounds $\chi \in [0.005, 0.04]$ and simulate several values within this range.

## Oxygenic photosynthesis in terrestrial photosynthetic niches

In Fig 3 we consider a selection of different terrestrial light environments calculated by filtering standard AM1.5 light from the previous section in different ways: redder light such as that found in the shade (top row) and bluer light as found under varying depths of clean water (middle and bottom rows). Further information about antenna composition and performance is given in Table 1 and Figs E–H in S1 Text. Each column in Fig 3 corresponds to

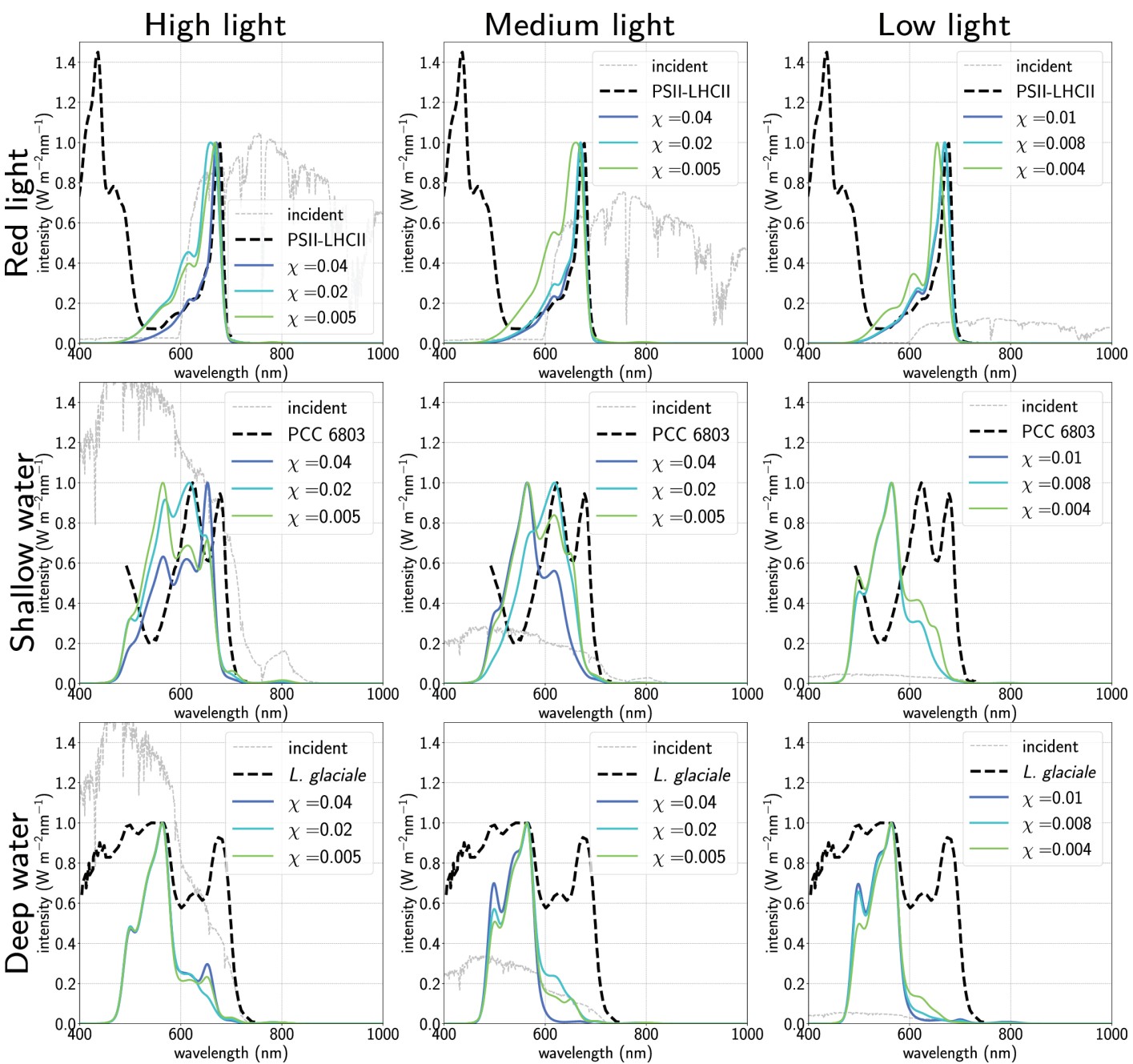

**Fig 3. Averaged absorption spectra as a function of cost $\chi$ for varying light environments (red light, top row; shallower water, middle row; deeper water, bottom row) and light levels (high intensity, left column; medium intensity, middle column; low intensity, right column).** The y-axis corresponds to the intensity of the incident spectra shown in grey. Dashed lines are experimental absorption spectra for, respectively, PSII-LHCII in spinach [55] (top row), cyanobacterium *Synecocystis* sp. PCC 6803 [56] (middle row) and marine red alga *Lithothamnion glaciale* [57] (bottom row).

a different overall intensity of light. The medium and low light simulations were performed by taking the high intensity filtered light and renormalising the integrated flux to 300 and 50 $\mu$mol m$^{-2}$ s$^{-1}$ of photosynthetically active radiation (400 - 700 nm, PAR) respectively. In

**Table 1. Details of antenna composition for different light environments and intensities, including branch number, $n_b$, and length, $n_s$, and approximate pigment composition.**

| Light | Light level | $n_b$ | $n_s$ | Pigment composition |
|---|---|---|---|---|
| Full sunlight | High | 3-4 | 3-5 | APC, PC, Chl $b$ |
| | Medium | 2-6 | 2-6 | APC, PC, Chl $b$ |
| | Low | 2-6 | 1-6 | APC, PC, Chl $a$ |
| Red light | High | 2-4 | 3-5 | Chl $a$, Chl $b$, APC |
| | Medium | 2-4 | 2-6 | Chl $a$, Chl $b$, APC |
| | Low | 3-5 | 2-6 | Chl $a$, Chl $b$ |
| Shallow water | High | 2-4 | 3-5 | APC, PC, r-PE, Chl $b$ |
| | Medium | 2-6 | 2-7 | PC, PE, r-PE, APC |
| | Low | 1-5 | 1-6 | PC, r-PE, APC |
| Deep water | High | 2-4 | 3-6 | PC, r-PE, PE, APC |
| | Medium | 2-6 | 2-7 | r-PE, PC, PE, APC |
| | Low | 1-7 | 1-7 | r-PE, PC, PE, APC |

each case our algorithm predicts structures using pigments commonly found in that environment as well as qualitatively reproducing important features of experimental absorption spectra.

We first simulate red light using a simple filter which blocks light of wavelengths shorter than roughly 600 nm [58], finding a population of antennae primarily composed of chlorophyll $a$ and $b$ as well as small amounts of allophycocyanin (APC), a cyanobacterial phycobiliprotein whose absorption profile is very similar to Chl $b$, especially for lower cost values. The resulting absorption spectra are shown in the top row of Fig 3 along with an experimentally measured absorption spectrum of the combined PSII-LHCII spectrum from spinach [55], showing good agreement in the $Q_y$ region of the spectrum.

We next simulate shallow ($\approx 1$ m depth) and deeper ($\approx 2.5$ m) water spectra using the formula for light attenuation by water given in [59]. Note that this is a crude approximation for a true aquatic spectral flux since real terrestrial water also contains varying concentrations of dissolved and particulate organic matter (known as gilvin and trypton in the literature), as well as phytoplankton, which absorb differently to the water itself. However, since the relative concentrations of these are heavily location-dependent on Earth, we choose to neglect them at this stage, meaning that the resulting spectral fluxes are generally bluer and more intense than would be expected at equivalent depth in real marine environments.

For the shallower water spectrum (middle row of Fig 3) we generally obtain antennae composed of APC, phycocyanin (PC) and different forms of phycoerythrin (PE), with small amounts of Chl $b$ at higher cost values. Varying the cost varies the proportions of these pigments, leading to shifts in the largest absorption peak from 650nm (APC) to 600nm (PC) to 550nm (PE). Additionally, reducing the light level also blueshifts the largest absorption peak by enriching the antennae with first PC and then PE. We note that specifically at low light and at cost $\chi = 0.01$ our algorithm cannot find any viable solutions: this seems to be due to the specific location of the peak in flux, and suggests that as the light intensity decreases and the cost increases the algorithm becomes increasingly sensitive to fine features in the input spectral flux.

In deeper water (bottom row of Fig 3 we see antennae enriched even further with PE, especially at higher cost values and lower light intensities, as our antennae evolve to capture as much flux in the blue region of the spectrum as possible. We note that some cyanobacteria also acclimate to bluer light by reducing their PC content in favour of PE via a process called type III complementary chromatic adaptation (CCA) [60,61], although they do not stop producing PC entirely. The extreme adaptations of the simulated antennae are possible since they

are free to completely overhaul their structure as necessary, and are therefore much more plastic (i.e. they can respond far more easily to the illumination they receive) than any real-world organism. Nonetheless, we do find absorption spectra with similar absorption maxima to real cyanobacteria in shallow water, and reproduce the blue absorption peak at ≈570 nm of certain species of red alga such as *Lithothamnion glaciale* which thrive in deep water with PE-enriched antennae [57].

Finally, we note that the average number of branches chosen by the algorithm does not seem very sensitive to light input, only to the cost parameter $\chi$. Naively, we might expect a larger number of very short branches to be favoured in general, especially where there is high flux at wavelengths close to that of the RC, but do not find this to be the case.

## Evolution of far-red adapted antennae

Certain species of cyanobacteria were recently shown to be able to acclimate to light environments which are extremely depleted in PAR but rich in Far Red (700 - 750 nm) light (often due to shading by other organisms) [58,63,64]. This response is known as FaRLiP (Far Red Light Photoacclimation) and involves the antenna and RC becoming enriched in chlorophyll *d* and/or *f*. Moreover, there is also evidence that the RC trap of PSII redshifts from its usual 680 nm to ∼ 720 nm, again via subsitution of different chlorophylls. We simulated a progressively PAR-depleted light environment by applying a sharper and sharper filter to reduce the amount of available visible light at wavelengths shorter 700 nm. For a non-FaRLiP RC with absorption at 680 nm, our algorithm has difficulty finding any solutions with positive fitness values, and hence a functional antenna does not evolve at all. This reflects the fact that absorbing light at > 700 nm and transferring energy uphill to a 680 nm RC is profoundly inefficient.

However, with a prototypical FaRLiP RC absorbing at 720 nm, we find that our antennae evolve successfully except at the very lowest PAR intensities and highest cost vales; we also see a transition from antennae predominantly composed of Chl *a* and *b* in Figs 4A and B to Chl *d* in Fig 4C as the amount of PAR decreases, matching the red absorption peak of the far-red adapted cyanobacteria *Acaryochloris marina* [62] at around 700 nm. At around 5% of the PAR intensity, for the highest simulated cost value, we see a change again from a Chl *d* antenna to a very small Chl *f* antenna; increasing the cost or decreasing the intensity further prevents the algorithm from being able to find any solutions at all. We also see in Fig 4D that there is a considerable decline in electron output as the main constituent of the antenna changes from Chl *a* and *b* to Chl *d* to Chl *f*, without a significant gain in efficiency; further information and details of pigment compositions can be found in Fig I in S1 Text. This is consistent with the fact that even FaRLiP cyanobacteria have restricted growth in Far Red light [65]. We attribute this to the fact that even though average antenna size decreases with increasing cost and decreasing visible light, moving from Chl *d* or Chl *f* to the reaction centre involves a smaller enthalpy decrease, and hence reduces the free energy gradient, leading to slower energy transfer between the two.

## Initial populations: contingent vs. *in-situ* evolution

So far all antenna have been assumed to evolve *in situ* from a random population of *proto-antennae*. This may miss potentially viable antenna configurations if no simple evolutionary path connects them to the proto-antennae. To investigate this, we take a population of "shallow water" antennae composed of APC and PC and let them *adapt* to "deep water" conditions (see Fig 5C and 5D), with further detail in Figs J and K in S1 Text). We find that our evolved antennae stay close in absorption profile to this initial population for each cost value,

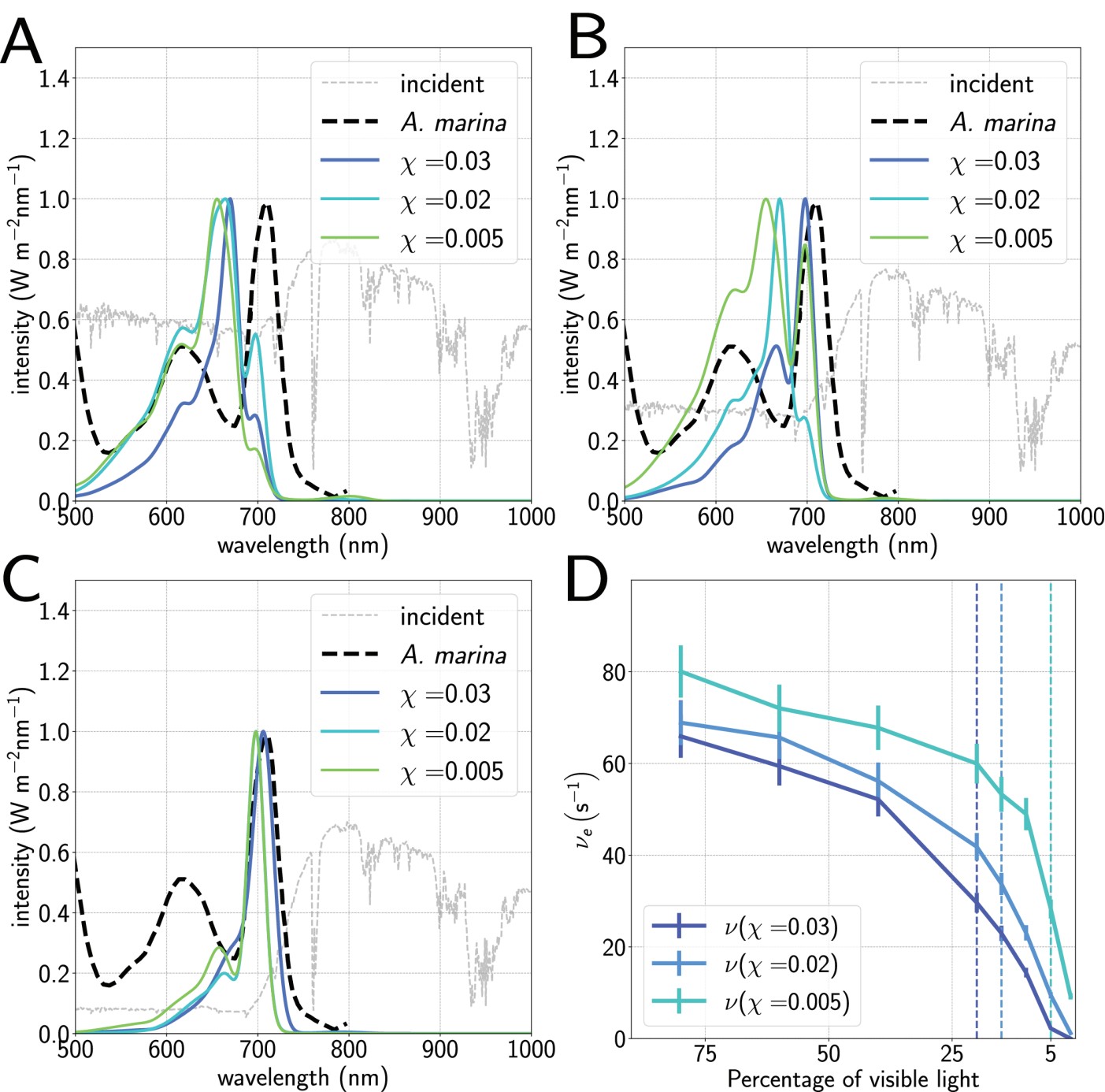

**Fig 4. Averaged absorption spectra as a function of cost compared to the far-red adapted cyanobacteria** *Acarychloris marina* **[62] for (A) 40% of available visible light below 700nm, (B) 20%, and (C) 5%.** A clear shifting of the main absorption peak as the intensity of visible light decreases is seen, corresponding to a shift from Chl *a* and *b* to Chl *d* in the antennae. (D) Electron output as a function of cost and percentage of available light under 700nm. Vertical dashed lines indicate approximately the fraction of shorter-wavelength light at which the antennae switch to using predominantly Chl *d* rather than Chls *a* or *b*, showing a significant decrease in electron output around this point.

with phycobilisome-like antennae composed of either APC or Chl *b*, PC and PE shown in Fig 5B, but in significantly different proportions compared to the *in-situ* evolution shown

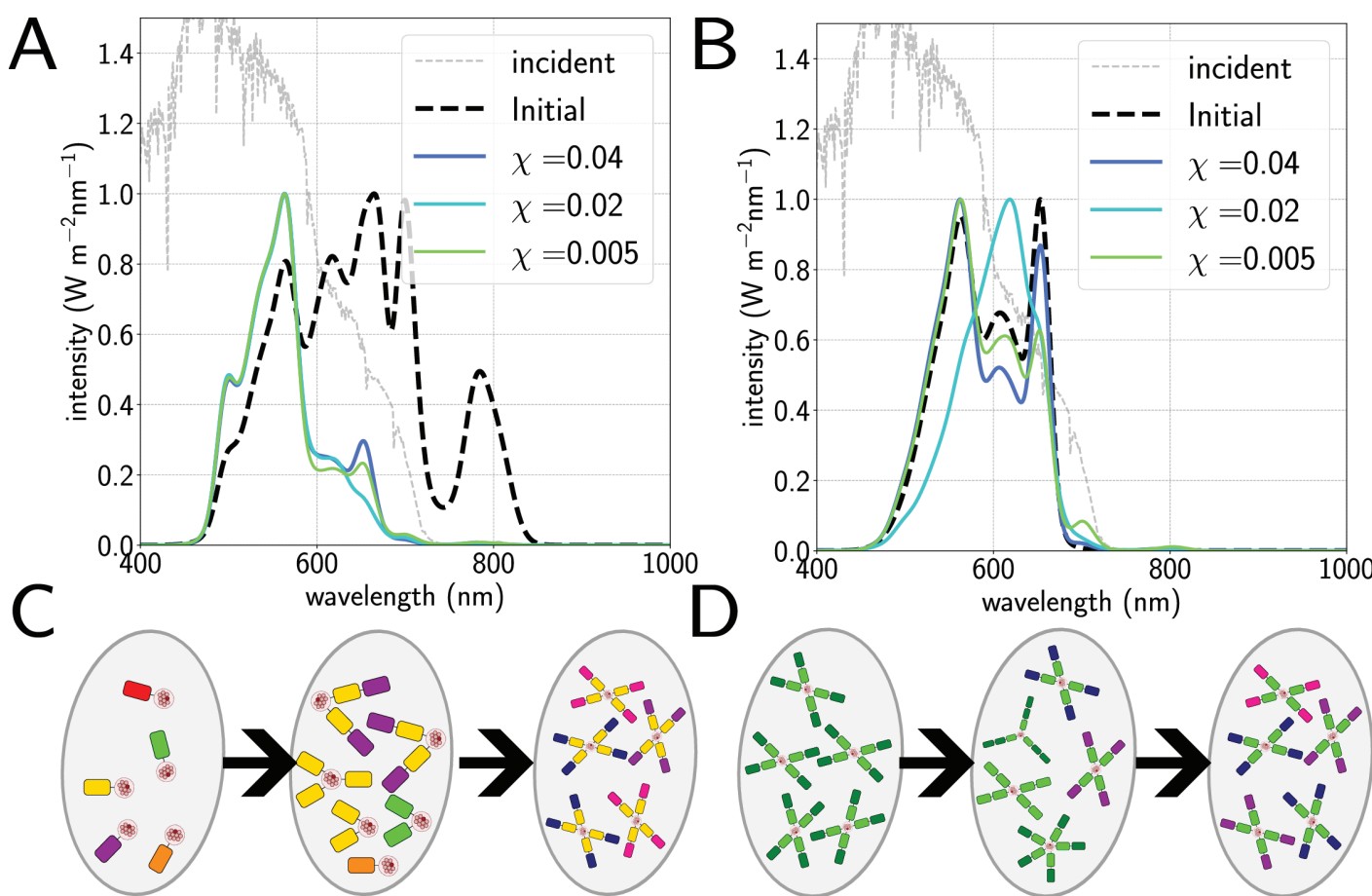

**Fig 5. Comparison of contingent versus *in situ* evolution of antennae under 2.5m depth of water.** A.) Average initial absorption spectrum for *in situ* evolution (solid lines), as a function of cost, compared to the initial spectrum of the unevolved population (dashed line). B.) Averaged final absorption spectra for the contingent (solid lines) populations as a function of cost compared to the initial spectrum (dashed line). C and D.) Diagrams illustrating evolution of *in situ* (C) and pre-adapted (D) populations from left to right, with different coloured blocks representing different pigment types (colours arbitrary).

in Fig 5A. The APC antenna core is generally either retained or replaced with Chl *b*, but not removed entirely, and forms a larger part of the overall absorption profile in the population pre-adapted to shallow water. In terms of antenna performance, $\nu_e$ and $\phi$ are slightly higher on average for the *in-situ* evolved population, but essentially identical within the error bounds for the two populations. This can be thought of more generally in terms of maxima of the "fitness surface": there may be multiple different antenna configurations which are roughly equal in performance for a given light environment, illustrating the history-dependence of evolutionary solutions [66].

## Oxygenic and anoxygenic photosynthesis in extra-solar light conditions

Given that our model appears qualitatively robust, we apply it to incident fluxes one would expect on the surface of exo-planets orbiting in the habitable zone of M- and K-dwarf stars. As in our previous work, these are generated by taking model spectral luminosities for various stellar temperatures [67], and assuming an orbital distance that would allow for liquid

water [7,68]. Since recent research has noted the potential feasibility of *anoxygenic* photosynthesis arising around cooler stars [69], we compare our oxygenic model to a crude anoxygenic model with a RC wavelength of 890 nm [70–72]. Fig 6 shows simulated absorption spectra and respective electron output for oxygenic and anoxygenic models for M- and K-dwarfs. We see that constrained by a 680 nm oxygenic RC, for the coolest M-dwarfs (top row), the antenna performance and even the location of the absorption peak varies significantly with cost–this is because lowering the cost allows larger antennae to evolve to capture more flux. Pigment compositions are detailed in Table. 2 and Figs L–N in S1 Text. The evolved population of antennae are composed mostly of Chl *b* at lower cost values, and exclusively Chl *d* at higher cost.

We note that even at the higher cost values, the low electron output does not necessarily mean oxygenic photosynthesis would be impossible: several species of cyanobacteria have adapted to very low levels of illumination [73], and simulated M-dwarf light does not limit growth of cyanobacteria, even without the FaRLiP response [6].

In contrast, for the warmer K-dwarfs (middle row) we see very reasonable antenna performance regardless of cost; interestingly, instead of exclusively Chl *b* antennae, we predict a phycobilisome-like composition of predominantly APC and PC with small amounts of chlorophyll mixed in.

For a 890 nm anoxygenic RC, under M-dwarf illumination, the model selects a larger antenna composed of BChl *b* with an absorption peak $\sim 800$ nm. The electron output is much higher ($\nu_e \geq 60 \text{ s}^{-1}$), which casually suggests that anoxygenic photosynthesis may be the better evolutionary choice, though numerous other factors will be important. For warmer stars (Fig 5B and 5C) we find antennae again predominantly composed of BChl *b*, but with BChl *a* and even Chl *d* included, as shown by the intermediate peak in the absorption spectra at around 700 nm.

Lastly we note that for the warmer K-dwarf stars ($T = 3800$ K) the performance of the oxygenic photosystem is roughly as good as that predicted for an anoxygenic photosystem, in agreement with recent research showing comparable growth for both vascular plants and desert cyanobacteria (both oxygenic organisms) under simulated K-dwarf light [74] to that in solar light.

## Discussion

Our model, which is *necessarily* coarse-grained, when coupled with a sophisticated evolutionary algorithm, seems to predict (qualitatively) the various oxygenic light-harvesting structures that have evolved in various Earth niches. This includes plant-like chlorophyll *a* and *b* antennae in redder light (typically of shaded environments) and cyanobacteria-like phycobilin-based antennae in bluer, sub-marine environments. Using a simple model of a far-red light adapted reaction centre, we show that the model predicts a shift from Chl *a* and *b* antennae to Chl *d* and *f* analogously to the FaRLiP response [20,63,64,75] as the intensity of visible light decreases and the available light becomes redder and redder.

Applied to the light from M-dwarf stars, it implies that oxygenic photosynthesis, with organisms whose antennae are chlorophyll-like, may still evolve, though with reduced output relative to Earth organisms. This may limit oxygenic photosynthesis to simpler, single-cell organisms that, unlike vascular plants, do not have to support large amounts of non-photosynthetic tissue.

We see these results as a promising first step in producing a universal model of photosynthesis. They suggest that we can, cautiously, hypothesize on the nature of biological

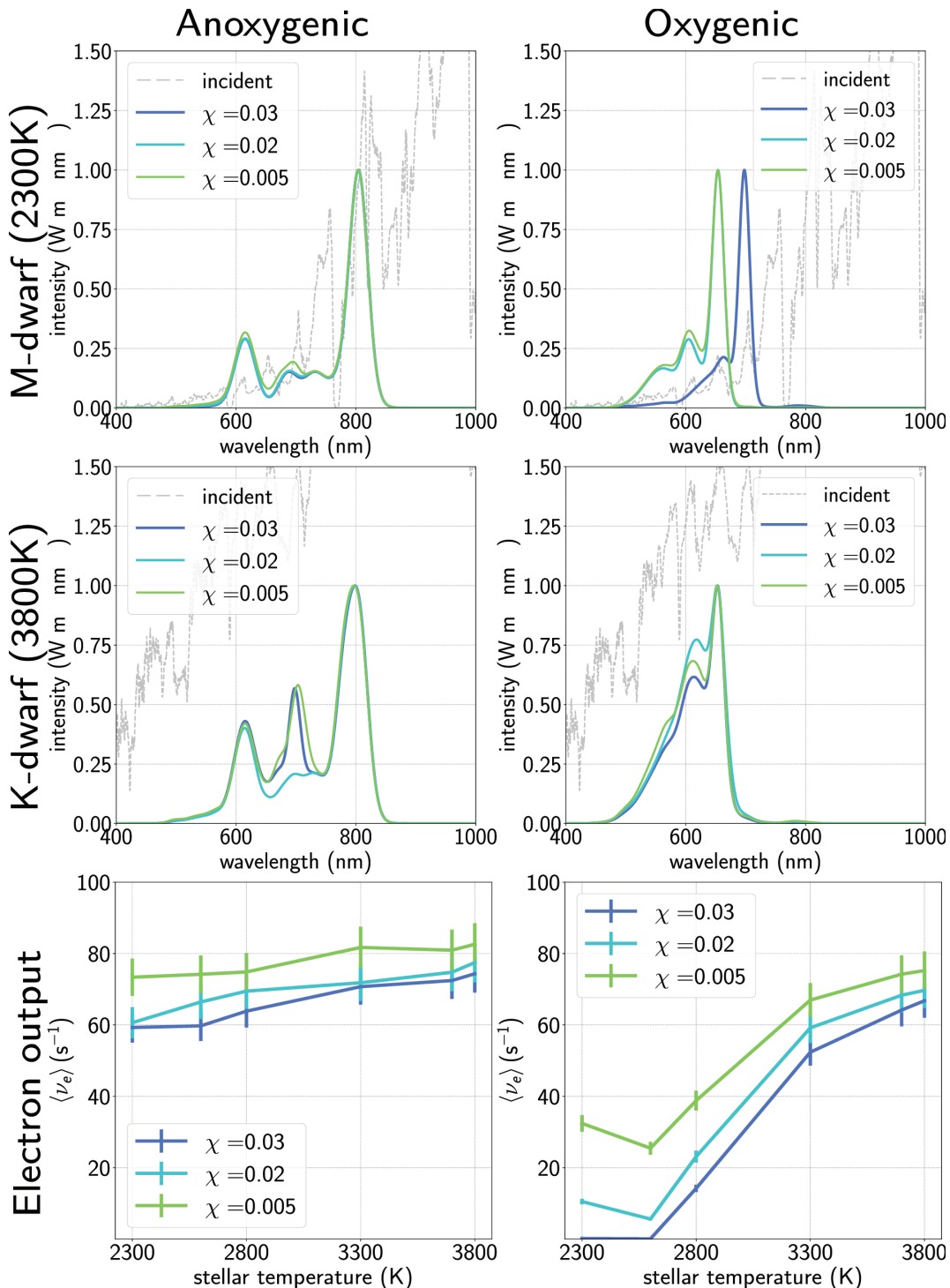

**Fig 6. Generated absorption spectra as a function of cost for cool M-dwarf (stellar temperature 2300K) (top row) and larger K-dwarf (stellar temperature 3800K) (middle row) for antennae with anoxygenic (left column) and oxygenic (right column) reaction centres respectively.** Our algorithm cannot produce viable oxygenic antennae at high cost values for the coolest M-dwarfs; by contrast the algorithm works well for anoxygenic photosynthesis for all cost values and stellar temperatures. Averaged electron output of the antennae are shown (bottom row) as a function of cost for various stellar temperatures, showing dramatically increasing performance of oxygenic photosynthesis as the stellar temperature increases compared to broadly similar performance for anoxygenic photosynthesis. For detailed pigment compositions see Figs L–N in S1 Text.

**Table 2. Details of antenna performance and pigment composition for anoxygenic and oxygenic photosynthesis as a function of stellar temperature.**

| Light | RC type | Electron output $(s^{-1})$ | Pigment composition |
|---|---|---|---|
| 2300K | Anoxygenic | 60-75 | BChl *b*, BChl *a* |
| | Oxygenic | $\approx$0-30 | Chl *b*, Chl *d* |
| 3800K | Anoxygenic | 70-80 | BChl *b*, BChl *a*, Chl *d* |
| | Oxygenic | 65-75 | APC, PC, Chl *b*, Chl *a* |

processes beyond Earth. That said, the model presented here does have certain shortcomings, consideration of which may help to direct subsequent work.

Firstly, the results obtained from the genetic algorithm seem relatively robust with respect to its main hyperparameter $\chi$, which for a given light environment serves primarily to constrain the size of the antenna and prevent pigments being added which capture no flux, rather than qualitatively altering the evolved structures. However, $\chi$ is a relatively broad and nebulously-defined quantity, and any further work focusing on more specific organisms or light environments may need refined strategies for constraining antenna size, for example by separating it into multiple separate parameters representing different costs. In addition, it does not account for the overall light intensity, and so the range of reasonable $\chi$ values decreases as the light intensity does; one obvious potential solution to this is to tie the cost to the integrated flux received by the antenna in some way. Organisms evolving in flux-limited environments may very well be rather finely-tuned, minimizing the overall cost of photosynthesis. How exactly such ideas should be incorporated into this model will require further, careful consideration.

Our genetic algorithm also allows for infinitely plastic behaviour–for example, an organism spontaneously starting to produce completely new kinds of pigments–which is biologically unlikely to say the least. Further work which focuses more specifically on the evolution of photosynthesis in specific terrestrial light environments would need to consider the mutation process more carefully in particular to address this.

Another limitation of the model as it stands concerns the relatively crude treatment of the RC. We use the simplest possible kinetics scheme of a two-step process involving fast primary charge separation, which reflects the transition from exciton transfer to charge transfer, followed by much slower 'electron output'. Of course this neglects the fact that the cycle of charge separation, acceptor reduction and donor oxidation is a multi-step process, on a hierarchy of timescales, that must compete with side processes such as *de-trapping* (reformation of the exciton) and charge recombination. We have focused here exclusively on light absorption and energy transfer by the antenna, as in previous works [34,36], as effective light-harvesting strategies are a near universal feature of photosynthesis and will be required if photoautotrophy is to evolve around M-dwarf stars. We have considered "standard" oxygenic RC energetics corresponding to PSII, a far-red adapted RC and an anoxygenic RC, but without explicitly considering the internal RC processes. A more quantitative description of the combined antenna-RC supersystem, for example to investigate the relative performance of anoxygenic vs. oxygenic photosynthesis or to examine the effect of redshifting the RC energy in more detail, is an obvious extension and will be the focus of further work.

## Supporting information

**S1 Text**

- **Table A.** Table of fit parameters for absorption spectra.

- **Table B.** Table of fit parameters for emission spectra.
- **Fig A.** Plots of experimental and fitted absorption and emission spectra for r-PE, PE, PC and APC.
- **Fig B.** Plots of experimental and fitted absorption and emission spectra for Chl *a*, Chl *b*, Chl *d*, Chl *f*.
- **Fig C.** Plots of experimental and fitted absorption and emission spectra for BChl *a*, BChl *b*.
- **Fig D.** Absorption spectra and pigment composition of antennae evolved under AM1.5 full sunlight.
- **Fig E.** Pigment composition of antennae evolved at high cost for terrestrial light environments.
- **Fig F.** Pigment composition of antennae evolved at medium cost for terrestrial light environments.
- **Fig G.** Pigment composition of antennae evolved at low cost for terrestrial light environments.
- **Fig H.** Electron output and efficiency antennae evolved in terrestrial light environments.
- **Fig I.** Pigment composition of antennae evolved with far-red adapted reaction centre.
- **Fig J.** Pigment composition of antennae evolved *in-situ* versus pre-adapted antennae.
- **Fig K.** Electron output and efficiency of antennae evolved *in-situ* versus pre-adapted antennae.
- **Fig L.** Pigment composition of antennae with anoxygenic and oxygenic reaction centres for M- and K- dwarf light at high cost values.
- **Fig M.** Pigment composition of antennae with anoxygenic and oxygenic reaction centres for M- and K- dwarf light at medium cost values.
- **Fig N.** Pigment composition of antennae with anoxygenic and oxygenic reaction centres for M- and K- dwarf light at low cost values.

(PDF)

## Author contributions

**Conceptualization:** Callum Gray, Samir Chitnavis, Tamara Buja, Christopher D. P. Duffy.

**Formal analysis:** Callum Gray, Christopher D. P. Duffy.

**Funding acquisition:** Christopher D. P. Duffy.

**Investigation:** Callum Gray, Samir Chitnavis, Tamara Buja, Christopher D. P. Duffy.

**Methodology:** Callum Gray, Samir Chitnavis, Tamara Buja, Christopher D. P. Duffy.

**Project administration:** Christopher D. P. Duffy.

**Resources:** Christopher D. P. Duffy.

**Software:** Callum Gray, Tamara Buja.

**Supervision:** Christopher D. P. Duffy.

**Validation:** Callum Gray.

**Visualization:** Callum Gray, Christopher D. P. Duffy.

**Writing – original draft:** Callum Gray, Samir Chitnavis, Tamara Buja, Christopher D. P. Duffy.

**Writing – review & editing:** Callum Gray, Samir Chitnavis, Christopher D. P. Duffy.

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
