## [Decision Letter · Decision Letter 0]

21 Oct 2024

Dear Dr. Gray,

Thank you very much for submitting your manuscript "Predicting the diversity of photosynthetic light-harvesting using thermodynamics and machine learning" for consideration at PLOS Computational Biology. As with all papers reviewed by the journal, your manuscript was reviewed by members of the editorial board and by several independent reviewers. The reviewers appreciated the attention to an important topic. Based on the reviews, we are likely to accept this manuscript for publication, providing that you modify the manuscript according to the review recommendations.

All three reviewers found the research presented in the manuscript to be an interesting study that will appeal to broad audiences on the how photosynthetic antenna could evolve to suit the light environment on other planets. There are few points raised by the reviewers of needed clarification in the manuscript that we ask you to address.

Sincerely,

Megan L. Matthews, Ph.D.

Academic Editor

PLOS Computational Biology

Marc Birtwistle

Section Editor

PLOS Computational Biology

All three reviewers found the research presented in the manuscript to be an interesting study that will appeal to broad audiences on the how photosynthetic antenna could evolve to suit the light environment on other planets. There are few points raised by the reviewers of needed clarification in the manuscript that we ask you to address.

Reviewer's Responses to Questions

**Comments to the Authors:**

Reviewer #1: The paper is very interesting and leverages years of research on natural light-harvesting in different ecological niches to open an unexplored, exciting research line.

I would recommend, for clarity, to add to the manuscript the following information:

-in the Intro, please explain for cooler and smaller M-dwarf stars what is the emission range, for clarity.

-line 88-89, please explain why being oxygenic antenna you choose specifically 5 ps and provide references

-lines 138-146: explain how your choice of identical branches differ from natural supercomplexes and how, choosing them identical, it might limit the model.

-Figure 3: the bottom plots are not clear, what are the different colors of the histograms referring to? also, I cannot find a comment in the text about this part of the figure. Also, and how does the distance from the RC from your models compare with the actual ones of natural supercomplexes? I cannot find it commented even in the text but I might have overlooked.

-the method references another method used by the author in a paper submitted for review, in the references. what is the difference between the two papers?

Reviewer #2: The manuscript by Gray et al. is an interesting exploration on the in silico evolvability of photosynthetic antenna structures for different solar spectra. The authors generated a coarse, but detailed enough, model for light harvesting and applied an evolutionary model to allow the systems to respond to the imposed light environments. Overall, the article is well written and of interest for the discussions arising from their results.

There are a few points of criticism/clarification that I belive would improve the manuscipt.

In general, all of the simulations were performed under 'high light'. Could the authors clarify what they mean by this in relative photon flux or other terms? Have simulations been performed at medium (say ~50% high light) and/or low (~5% of high light)? That would be interesting to compare the pigment composition, as well as pigment and subunit number of the antenna.

-Line 11: I do not belive that the parent start is the most obvious source of energy for life. One only needs to look at Earth, where LUCA would not have been capable of photosynthesis (see https://doi.org/10.1038/s41559-024-02461-1 for a recent estimate of ~4.2 Ga for LUCA versus https://doi.org/10.1111/gbi.12322 generating an estimate of ~3.4 Ga for the origin of the Type II reaction center proteins). This does not change the relevance of the manuscript, but does greatly alter the importance of photosynthesis for establishing life on a planet.

Lines 17-18: The Em for the half reaction of water oxidation is +0.82 meV at pH 7.0, while P680+ has a potential of ~1.2 eV which provides an over-potential for water oxidation. Shifting the wavelength of P to 727 nm as in the chlorophyll f PSII still results in an overpotential for water oxidation. The overpotential may be necessary for the biological dissipation processes within the reaction center itself (not considered in this work), however, red shifting the reaction center potential to still achieve water oxidation is fully feasible. It may be worthwhile for the authors to include these potentials to better highlight their point.

Lines 49-50: For most antenna, describing them as energy funnels is not particularly true. This is likely only the case for phcobilisomes which clearly act as funnels from higher to lower energy pigments. In most antenna, however, the pigments are near isoenergetic with dozens (or hundreds) of pigments within a few nanometers of identical absorption.

Line 55-57: "Lastly, the antenna should not be so large and complex that its creation and maintenance leaves little surplus energy for growth, reproduction, etc." I would suggest the authors clarify if large and complex are separate entities or one. For example chlorosomes are large (hundreds of thousands of chlorophyll molecules) but one could argue not complex.

Lines 83-84: Why were RC dissipative processes neglected? If modeled are the results the qualitatively the same? Or does it become too cumbersome to model? Please clarify.

Bottom of page 4 (should be lines 85-87 but are not numbered): "Open" and "Closed" are commonly used terms to describe the reduction state of reaction centers. it seems unnecessary to use alternative nomenclature.

Figure 1: Figure 1 is not referenced in the text at all except for 1B. The ordering of the panels is also slightly confusing and could be shuffled to make the current letters A-E as: A, B (top), C (middle), D, E (bottom) and maintain alphabetic ordering left to right rather than top to bottom then top to bottom again.

Figure 1B: These are the absorption spectra of pigments in solvent, presumably? Please clarify.

Line 86-87: The rate of trapping varies for open vs closed PSII. Please explain where the 5 ps trapping time came from, as the cited paper (Broess et al.) appears to only reference 5 ps as a kinetic needed to account for the excitation light. Most reported PSII trapping rates are 2 orders of magnitude larger.

Line 197: please correct (ref)

For all figures containing solar spectra and results, please scale the incident solar spectra to be visible in the panel. This allows some understanding and comparision between different situations. See Fig. 2C, Fig. 5B

For Fig. 2C, is the intensity (y-axis) refering to the absorption spectra or the indicent light? This could be made clearer on all figures with multiple parameters plotted.

Many of the supplemental figures are not referenced in the text of the manuscript and should be.

Figure 3: The dotted spectra represent different complexes from PSII, phycobilisomes, and whole cells. However, many of the individual (PSII) and combined (PSII and phycobilisome) spectra are available, especially rather than using whole cell spectra. Clarifying these spectra could help, as well as clarifying that the authors are showing the similarity (or lack thereof) to antenna rather than the antenna-PSII complex. Please also scale the dashed line in D to fit on the graph.

Line 278: Type III is gererally used for complementary chromatic acclimation, not group III

Figure 5: It would be useful to also display the antenna composition for the anoxygenic antennas

Line 332: '...utilising a Chl a antenna' I think would be better phrased that on other planes an organism could exists using antenna with similar spectral properties as chlorophyll a.

The far-red light evolution is interesting for its lack of evolving. However, in these organisms the phycobilisomes are also still present though wavelength shifted, with altered PSII trapping rates as well (see Mascoli et al. Nature Communications 2022). If the trapping rates are altered in this environment, can an antenna evolve? For non-experts, it would help if the authors described how their results show that the system did not evolve.

It is interesting that chlorophyll d is so rarely represented in the antenna. It seems to only be prevalent in far-red light when the cost is low. However exclusively chlorophyll d antenna are found in Acaryochloris species. If the reation center is shifted to 727 nm does chlorophyll d become used in any radiation spectrum?

Reviewer #3: The paper by Gray et al. presents a fascinating and exciting study of the possible biodiversity of photosynthesis on exoplanets hosting different environments than Earth. The authors construct a physical model of photosynthesis based on the known physics of photosynthetic antennae on Earth and simulate the evolution of photosynthetic antennae under different illumination conditions and different costs of antenna production using a genetic algorithm. Interestingly, they reproduce roughly and qualitatively the properties of photosynthetic antennae in various environments on Earth, which gives them the confidence to attempt predictions for extra-terrestrial environments.

The physical model the authors use for photosynthesis is a solid one, and it probably works because the most important steps of energy transfer in photosynthesis are based on weak coupling between pigments or whole antennae. It is, therefore, possible to estimate the rate by the formula (4) of the paper. The authors present a general model of exciton transfer and restrict it to a single excitation regime valid for photosynthesis on Earth. They find a range of production costs that gives them reasonable results - i.e. non-trivial antennae and with this range they study the conditions characteristic of exoplanet environments.

The study is a very valuable one because it provides physical backing to hypotheses about photosynthesis evolution on exoplanets. It is easy to see that this study is open to many ways of generalization and modification as it provides just a first step in what may become a fields of its own. In fact, an evolutionary study like this can provide some sanity check to speculations about the optimization of photosynthesis by tuning various details of the antenna which are often suggested by theoretical studies. Here the model is simple, robust, and void of weird quantum effects, yet it captures essential characteristics of real photosynthesis. I wholeheartedly recommend this paper for publication.

I have only one question, which the authors may use to optionally revise their manuscript. I was missing the information about the production cost for the existing organisms. It would boost the confidence of the reader if this number agreed with the simulation results.

**Have the authors made all data and (if applicable) computational code underlying the findings in their manuscript fully available?**

Reviewer #1: None

Reviewer #2: **No: **The github link to the code is not active. Data availability on dryad is unclear and a location not stated

Reviewer #3: Yes

PLOS authors have the option to publish the peer review history of their article (what does this mean?). If published, this will include your full peer review and any attached files.

Reviewer #1: No

Reviewer #2: No

Reviewer #3: No

Figure Files:

Data Requirements:

Reproducibility:

References:

---

## [Decision Letter · Decision Letter 1]

3 Feb 2025

Dear Dr. Gray,

We are pleased to inform you that your manuscript 'Predicting the diversity of photosynthetic light-harvesting using thermodynamics and machine learning' has been provisionally accepted for publication in PLOS Computational Biology.

Best regards,

Megan L. Matthews, Ph.D.

Academic Editor

PLOS Computational Biology

Marc Birtwistle

Section Editor

PLOS Computational Biology

Reviewer's Responses to Questions

**Comments to the Authors:**

Reviewer #2: The revised manuscript by Gray et al. improves their originally submitted manuscript in multiple ways as described by the authors in their response letter. The inclusion of further simulations, combined with a structured results section that, in my opinion, is more clearly organized greatly improves their paper.

Overall, the manuscript has a clear, reasonable objective which is tested under multiple circumstances. These are then well described, compared to one another and nature, and discussed thoughtfully.

This work will be of interest to multiple research fields.

There are a few minor features that could be fixed by the authors:

Line 29: 'life, life, wherever...' should be fixed

Line 112: contain should be containing or 'and contain'

Line 153 should refer to 1D not 1B

Figure 1E: what does the grey line represent?

Line 161: 'rae' should be 'rate'

Line 169: -1 should be superscripted

Line 196: sever should be severe

Line 505: 'We simulated a progressively Far Red this by applying...' this should be removed/fixed

Line 507: FarLiP should be FaRLiP

Line 631: 'and will be required doubly-required' should be fixed

**Have the authors made all data and (if applicable) computational code underlying the findings in their manuscript fully available?**

Reviewer #2: Yes

PLOS authors have the option to publish the peer review history of their article (what does this mean?). If published, this will include your full peer review and any attached files.

Reviewer #2: No

---

## [Editor Report · Acceptance letter]

PCOMPBIOL-D-24-01295R1

Predicting the diversity of photosynthetic light-harvesting using thermodynamics and machine learning

Dear Dr Gray,

I am pleased to inform you that your manuscript has been formally accepted for publication in PLOS Computational Biology. Your manuscript is now with our production department and you will be notified of the publication date in due course.

With kind regards,

Zsofia Freund
